Microbiota based personalized nutrition improves hyperglycaemia and hypertension parameters and reduces inflammation: a prospective, open label, controlled, randomized, comparative, proof of concept study

Kallapura Gopalakrishna 1
Prakash Anthony Surya 1
Sankaran Kumar 1
Manjappa Prabhath 1
Chaudhary Prayagraj 2
Ambhore Sanjay 3
http://orcid.org/0000-0003-1448-4002 Dhar Debojyoti 1 ddhar@leucinerichbio.com
1 Leucine Rich Bio Pvt Ltd , Bengaluru , India
2 IIT (BHU) , Varanasi , India
3 Shreya Clinic , Bhopal , India
Menini Stefano
Electronic publication date: 2024 Jun 26
Publication date: 2024
Volume: 12
Electronic Location ID: e17583
Received 2024 Jan 19; Accepted 2024 May 27
Copyright: © 2024 Kallapura et al.
Copyright year: 2024
Copyright holder: Kallapura et al.
License: This is an open access article distributed under the terms of the Creative Commons Attribution License, which permits unrestricted use, distribution, reproduction and adaptation in any medium and for any purpose provided that it is properly attributed. For attribution, the original author(s), title, publication source (PeerJ) and either DOI or URL of the article must be cited.
License URL: https://creativecommons.org/licenses/by/4.0/

Keywords: HbA1c, Gut microbiome, Hyperglycemia, Type 2 diabetes, Personalized nutrition

Funding: Leucine Rich Bio Pvt Ltd This work was funded by Leucine Rich Bio Pvt Ltd. The funders had no role in study design, data collection and analysis, decision to publish, or preparation of the manuscript.

==============================
Background

Recent studies suggest that gut microbiota composition, abundance and diversity can influence many chronic diseases such as type 2 diabetes. Modulating gut microbiota through targeted nutrition can provide beneficial effects leading to the concept of personalized nutrition for health improvement. In this prospective clinical trial, we evaluated the impact of a microbiome-based targeted personalized diet on hyperglycaemic and hyperlipidaemic individuals. Specifically, BugSpeaks®-a microbiome profile test that profiles microbiota using next generation sequencing and provides personalized nutritional recommendation based on the individual microbiota profile was evaluated.

Methods

A total of 30 participants with type 2 diabetes and hyperlipidaemia were recruited for this study. The microbiome profile of the 15 participants (test arm) was evaluated using whole genome shotgun metagenomics and personalized nutritional recommendations based on their microbiota profile were provided. The remaining 15 participants (control arm) were provided with diabetic nutritional guidance for 3 months. Clinical and anthropometric parameters such as HbA1c, systolic/diastolic pressure, c-reactive protein levels and microbiota composition were measured and compared during the study.

Results

The test arm (microbiome-based nutrition) showed a statistically significant decrease in HbA1c level from 8.30 (95% confidence interval (CI), [7.74–8.85]) to 6.67 (95% CI [6.2–7.05]), p < 0.001 after 90 days. The test arm also showed a 5% decline in the systolic pressure whereas the control arm showed a 7% increase. Incidentally, a sub-cohort of the test arm of patients with >130 mm Hg systolic pressure showed a statistically significant decrease of systolic pressure by 14%. Interestingly, CRP level was also found to drop by 19.5%. Alpha diversity measures showed a significant increase in Shannon diversity measure (p < 0.05), after the microbiome-based personalized dietary intervention. The intervention led to a minimum two-fold (Log2 fold change increase in species like Phascolarctobacterium succinatutens, Bifidobacterium angulatum, and Levilactobacillus brevis which might have a beneficial role in the current context and a similar decrease in species like Alistipes finegoldii, and Sutterella faecalis which have been earlier shown to have some negative effects in the host. Overall, the study indicated a net positive impact of the microbiota based personalized dietary regime on the gut microbiome and correlated clinical parameters.

Introduction

The human gut microbiome, a complex ecosystem of trillions of microorganisms, plays a crucial role in our health and disease. It does so by influencing various physiological processes, including metabolism, nutrition, immunity, and even cognitive and behavioural functions (Kim et al., 2019; He & You, 2020; Cunningham, Stephens & Harris, 2021; Vandeputte, 2020; Gomaa, 2020). Further, the gut microbiome’s composition and diversity vary among individuals, influenced by factors such as age, diet, and environment (Odamaki et al., 2016). A balanced gut microbiome or eubiosis, is crucial for health, while an imbalance or dysbiosis, can contribute to various diseases like systemic inflammation, insulin resistance, and autoimmune and metabolic disorders (Srivastava et al., 2022). Furthermore, a strong and expanding evidence base supports the influence of gut microbiota in human metabolism, particularly in relation to conditions like hyperglycaemia and hyperlipidaemia (Kim et al., 2019; He & You, 2020; Cunningham, Stephens & Harris, 2021; Srivastava et al., 2022).

Hyperglycaemia, characterized by high blood sugar levels (Kim et al., 2019; Cunningham, Stephens & Harris, 2021), and hyperlipidaemia, marked by high levels of lipids in the blood (Kim et al., 2019; He & You, 2020; Brunkwall & Orho-Melander, 2017), are both metabolic disorders often associated with type 2 diabetes mellitus (T2DM) and cardiovascular diseases, respectively (Kim et al., 2019; He & You, 2020; Cunningham, Stephens & Harris, 2021; Srivastava et al., 2022; Brunkwall & Orho-Melander, 2017). Altered glucose homeostasis is associated with altered gut microbiota, which in turn is clearly associated with the development of T2DM and associated complications, by increasing serum concentrations of branched-chain amino acids causing insulin resistance (Cunningham, Stephens & Harris, 2021; Brunkwall & Orho-Melander, 2017; Martínez-López et al., 2022). For instance, a study conducted in 2019 found that the relative abundance of several bacterial taxa was significantly higher in individuals with diabetes plus hyperlipidaemia, with several bacteria correlated with fasting plasma glucose and blood lipid levels of the participants (Martínez-López et al., 2022; Liu et al., 2019).

Emerging research have consistently demonstrated that the gut microbiota can impact the nutritional status and health of the host. The gut microbiota has been shown to influence control over essential processes such as nutrient absorption, storage, and metabolism. These findings suggest the potential role played by the gut microbiota, including its metabolites, which can be possibly harnessed to promote regulation of metabolic syndrome (Vandeputte, 2020; Gomaa, 2020; Brunkwall & Orho-Melander, 2017; Martínez-López et al., 2022). Specifically, this can be achieved by exploiting the responsiveness of gut microbiome to changes in diet. In other words, the gut microbiome has been shown to adapt remarkably fast to alterations in our diet, leading to differential abundance of various bacteria and their diversity based on the type of food consumed. This adaptability can be advantageous for positive regulation of metabolism (Leeming et al., 2019). Consequently, modulation of the gut microbiota, through various means, has emerged as an important tool to fulfil nutritional requirements, combating malnutrition and even diseases such as hyperglycaemia and hyperlipidaemia (Kim et al., 2019; He & You, 2020; Liu et al., 2019).

Several recent studies have reported the application of personalized nutrition to improve intestinal microflora, and in turn the health status of the individual (Vandeputte, 2020; Lee, Davies & Barnett, 2023; Valdes et al., 2018; Song & Shin, 2022). Specifically, the difference in individual responsiveness based on the gut microbiota has the potential to become an important research approach for personalized nutrition and health management (Valdes et al., 2018). In the context of hyperglycaemia, the gut microbiota’s influence on postprandial glycaemic responses to identical meals has been demonstrated (Wilson et al., 2021). This suggests that a personalized diet based on one’s gut microbiome could significantly help in lowering hyperglycaemia and alleviating its negative effects (Singh et al., 2017; Ben-Yacov et al., 2023). A study conducted by the Mayo Clinic Centre for individualized medicine found that a personalized diet based on one’s microbiome (along with genetics, age, and activity level) is a far better way to control one’s blood glucose than cutting carbohydrates and calories (Mendes-Soares et al., 2019; Stanimirovic et al., 2022; Kallapura et al., 2023). This approach fits into precision medicine paradigm by considering different diet patterns and adopting the best one based on individual microbiota composition to achieve significant adiposity reduction and improve metabolic status (De Coster et al., 2018; Nurk et al., 2022; Langmead & Salzberg, 2012).

Therefore, microbiota modulation through diet provides an impactful tool to improve disease conditions such as type 2 diabetes. In this context we have evaluated BugSpeaks® which is a microbiome profiling test that provides nutritional recommendation based on the individual’s microbiota profile ascertained through whole genome shotgun metagenomics sequencing. BugSpeaks® derived personalized nutritional recommendation is individualized and aims to provide individualized nutritional recommendation to improve balance in the gut microbiota ecosystem. It is theorized that improved balance in the gut microbiota ecosystem can decrease chronic inflammation, help improve production of SCFA and other helpful metabolites thus alleviating chronic disease symptoms and improving overall health. In this prospective interventional trial therefore, we have investigated the impact of BugSpeaks® in hyperglycemic individuals. Specifically, the impact on clinical parameters such as HbA1c, total cholesterol, LDL, HDL, triglycerides, non-HDL cholesterol, CRP and IL-10 (inflammatory markers) have been evaluated. HbA1c, dyslipidaemia parameters and inflammation markers such as CRP have been evaluated in type 2 diabetes patients earlier (Schofield et al., 2016; Stanimirovic et al., 2022). Most importantly, the impact of the personalized diet on the gut microbiome, and possible correlations with the clinical parameters have also been evaluated.

Materials and Methods

Portions of this text were previously published as part of a preprint (Kallapura et al., 2023)

Study design and subjects’ selection

This prospective interventional trial was an open label, controlled, randomized, comparative, parallel group study, with two arms, conducted in conformity with ICH-GCP (E6 R2) guidelines, the Helsinki Declaration, and the local regulatory requirements (Indian GCP, Indian Council of Medical Research, and New Drugs and Clinical Trials Rules-2019). There was no further changes or amendments made after protocol approval. The study was initiated only after the receipt of ethics committee (EC) approval (Institutional Ethics committee, Charak Hospital- Reg:ECR/152/Inst/MP/2021, Bhopal, India). After obtaining the informed consent, subjects were screened by undergoing various assessments as per the schedule of assessment mentioned in protocol. This trial is registered and approved with the clinical trials registry—India with the following number CTRI/2022/05/042791 on 24/05/2022.

The trial included 30 Indian adults with hyperglycaemia and hyperlipidaemia, with HbA1c ≥ 8 or LDL cholesterol ≧120 mg/mL, or both. Both male and female participants were included in the study, with an age range of 42–65 years (Mean 53.82 ± SD 7.97 yrs.); with varying body weights (Body Mass Index (BMI) of 19.6–33.3 kg/m2, mean 24.32 ± SD 2.89); willingness to provide written informed consent and comply with study instructions for its duration, specifically agree to follow a personalised diet for 3 months. Exclusion criteria included subjects with history of alcohol, smoking or tobacco consumption; history of clinically significant physiological or neurological or psychiatric disease; organ transplantation or surgery in the past 6 months; known hypersensitivity or idiosyncratic reaction or intolerance to any dietary changes or any related products as well as severe hypersensitivity reactions (like angioedema) to any drugs or food products; difficulty with donating blood were excluded from the study. With the evaluation of gut microbiome in mind, individuals treated with oral antibiotics during 2 weeks prior to the study, who are undergoing any dietary restrictions, who consume antioxidant supplements, fermented foods (>3 servings per week) and/or laxatives, were also excluded from the study. Women who consume oral contraceptives or who are pregnant or breastfeeding, were also excluded from the study. Participants who met the necessary inclusion criteria were further encouraged not to change their current physical activities, and to refrain from any changes in their dietary habits before starting the clinical trial.

After obtaining the informed consent, subjects were screened by undergoing various assessments. Subsequently, eligible subjects were randomized to receive either BugSpeaks® based personalised diet (15 subjects) or regular diet (15 subjects) for 90 days at the trial site. Subjects in both arms continued with their stable dose of diabetic medication (Sulphonylureas, DDP4 Inhibitors, Thiazolidinediones, Nateglinide). The dietary restriction in the study were monitored for 90 days under the supervision of investigator through four onsite visits, Day 1, Day 30, Day 60 and Day 90 and daily patient diary recording. Briefly, the subjects were provided nutritional recommendations and meal plans. The subjects were asked to follow the meal plan and self report the food items consumed in a patient diary. This was remotely monitored by the investigator and nutritionist by phone call and during onsite visits. Safety assessment was done through subject reporting and laboratory parameters.

Investigation product: BugSpeaks®

The investigation product (IP) used in this clinical trial was a personalized gut microbiome-based diet, generated based on the individual’s gut microbiome. Briefly, the personalization of the diet was based on an in-silico compilation of associations between gut microbiome, microbial metabolism, disease and nutrition (and in extension foods). We overlayed and integrated these resources on to an individual’s gut microbiome profile in order to establish nutritional associations between one’s gut microbiome, with the overall objective of formulating a personalized set of dietary recommendations. To elaborate, we have created a proprietary database integrating current knowledge of gut microbiome, microbial metabolism and nutrition (and in extension foods). The database is an effort to interlink and integrate these resources to an individual’s microbiome, with the objective of formulating a set of dietary recommendations personalized for the individual. Basically, the microbe abundance information is fed into a proprietary algorithm that considers the information of which food item is associated with either increasing or decreasing its abundance (derived from a curated proprietary database). Depending on the overall abundance profile the final frequency of the food items are arrived at. This leads to the generation of a personalized nutritional recommendation based on the microbiota profile of an individual. In this study, the characterized microbiome of every participant in the intervention group was overlayed with this curated information, to generate personalized dietary recommendations, using proprietary algorithms. Largely, the objective is to customize the diet, with different frequencies of foods, in order to increase largely beneficial microorganisms and reduce any dysbiosis in the gut.

Study protocol and intervention

The current randomized prospective study was conducted as per the schedule provided in Fig. 1 (study design). The trial initiated with screening and baseline testing of all the subjects followed by stool sample collection for gut microbiome testing. Block randomization of 15 participants to either test arm (receiving BugSpeaks® gut microbiome-based personalized diet) or control arm (receiving regular diet) for the random sequence for treatment allocation was generated using online randomization tool (https://ctrandomization.cancer.gov/). The gut microbiome was profiled for the 15 participants of the test arm using whole genome shotgun metagenomics and personalized diet regimes were generated based on the individual microbiota profile. Personalized diet plans based on the individual’s microbiota profile were generated using algorithms and matrices which took into consideration abundance of various microbes and the effect of various food items in modulating their level.

Figure 1 Study design.

A flow chart depicting the study design, with two arms of the study, list of clinical parameters evaluated as primary end points and the microbiome profiling for the intervention arm (left).

During the intervention period (day 1 to 90), all the participants were instructed to follow either the BugSpeaks® gut microbiota based personalized nutritional meal plan (test arm) or regular diet regimes (diabetic meal plan—control arm) under the supervision of a dietician and principal investigator. Under the diabetic meal plan the participants were provided tailored meal plans with focus on food items that are recommended for patients with type 2 diabetes. The daily diet and occurrence of adverse events were recorded throughout the trial period. Site visit was planned on day 1, 30, 60 and 90 of the study periods. HbA1c, CRP, IL-10, triglycerides, LDL and HDL and anthropomorphic parameters such as systolic and diastolic blood pressure, BMI etc., were evaluated for all the participants on Day 1 and at the end of the study on day 90. One participant in the control arm was lost to follow up. Faecal samples of participants of the test arm were collected for microbiome sequencing, analysis and for providing nutritional recommendation based on the microbiota profile of each of the participants.

Plasma parameters

Plasma concentration of total cholesterol, LDL, HDL, triglycerides, non-HDL cholesterol, were estimated using photometric methods. While HbA1c was measured by HPLC, CRP was measured using immunoturbidimetry and IL-10 was determined employing ELISA. All plasma parameters were measured at Samadhan Pathology and Diagnostics, Bhopal.

Statistical analysis

A preliminary Shapiro-Wilk test for normality was conducted on a significance level of 0.05, using NumPy-Matplotlib (V.1.5.0) function within Python (V.3.11.8). The “shapiro” function in “scipy.stats” was used to calculate the p-value of the Shapiro-Wilk test. The Shapiro-Wilk test static was estimated along with p-value. Since the p-values were > 0.05 the data points were deemed approximately normally distributed. Later, the difference between test and placebo group was statically analysed using student’s t test. All the data was represented as mean ± standard deviation (SD). A p value < 0.05 was denotes statistical significance unless specified. Also, all endpoints were analysed separately, however gut metagenomic analysis was performed with different suite of tools, as described below.

Gut microbiota analysis

Faecal samples subjects in the test arm were collected 7 days before intervention (day 1) and on the last of the study period (Day 90). Gut microbiota of all the subjects belonging to the test arm were processed, sequenced and analysed at Leucine Rich Bio Pvt Ltd., India, using shotgun metagenome sequencing method, as detailed below.

Sample collection

Stool samples were collected from participants in the test arm using Invitek Molecular Stool Collection Module (Cat. No. 1038111300, Berlin, Invitek Molecular GmbH). Each participant was given the stool collection kit, with clear instructions about sample collection. The stool collection tube contained 8ml of DNA stabilizing solution and an integrated spoon in cap. All participants were instructed to collect ~2–3 spoons of stool into the 8ml stabilizing solution. Once collected, they were instructed to gently mix the sample with the stabilizing solution for 15 s, seal and then shipped under room temperature to the processing unit for DNA extraction.

DNA extraction

DNA was extracted from stool samples using QIAamp® Fast DNA Stool Mini (Cat No./ID; 51604; Qiagen, Hilden, Germany) following the manufacturer’s “Fast DNA Stool Mini Handbook” for fast purification of genomic DNA. Briefly, the extraction protocol consisted of two major steps: Lysis of and separation of impurities from stool samples and purification of DNA thereafter. Lysis of and separation of impurities from stool samples was carried out using the InhibitEX Buffer (Cat No./ID: 19593; Qiagen, Hilden, Germany), during which cellular structures release their DNA content in the solution. The sample matrix was pelleted by centrifugation and the DNA in the supernatant was purified on QIAamp Mini spin columns, which involved removal of proteins, binding DNA to the QIAamp silica membrane, washing away impurities, and eluting pure DNA from the spin column. Eluted DNA was collected in 1.5 ml DNA Lo-Bind microcentrifuge tubes, and the quantity and quality were assessed by Qubit 2.0 DNA HS Assay (ThermoFisher, Waltham, Massachusetts, USA) and NanoDrop® (Roche, Basel, Switzerland, USA) to meet the sequencing requirements.

Sequencing

Whole metagenome sequencing was performed on all samples using long read sequencing technology. Briefly, the DNA library was prepared with the Ligation sequencing kit (SQK-LSK114) (Oxford Nanopore Technologies (ONT), Oxford, UK), then loaded onto a R10.4.1 MinION flow cell (FLO-MIN114) and sequenced on the ONT MinION Mk1C device (MIN-101C). Basecalling and demultiplexing of sequence reads was performed with Guppy v4.2.2 and with assistance by MinKNOW GUI v20.10. Raw sequencing reads were stored in FastQ format for further computational analysis.

Upstream analysis

The upstream analysis involved quality check and quality improvement measures, including but not limited to host (human) sequence removal. This was followed by alignment of quality processed reads to a reference database of microbial genomes. The % normalized abundances, of all the microorganisms identified within these samples, were quantified, and later used for downstream analysis involving various statistical measures.

To elaborate, a thorough quality check of the raw sequencing data and some quality improvement measures were adopted to retain only quality reads for further processing. Primarily, the pre-processing operations included a quality check through NanoStat (De Coster et al., 2018) (v1.4.0) (https://github.com/wdecoster/nanostat) and removal of short and sub-par quality reads. Later, the reads that were deemed suitable for further analyses were mapped to the latest stable version of the human reference genome GH38 (Nurk et al., 2022) (GRCh38), using Bowtie2 (Langmead & Salzberg, 2012) (v2.5.2), to align and filtered out host (human) sequences from the data.

Kraken 2, a taxonomic classification system that uses exact k-mer matches to achieve high accuracy and fast classification of sequences was utilized for rapid, accurate, and sensitive microbial classification and quantification of species within the samples (Wood, Lu & Langmead, 2019) (https://github.com/DerrickWood/kraken2/wiki/About-Kraken-2). A custom database, built on the comprehensive, integrated, non-redundant, well-annotated set of sequences from Reference Sequence (RefSeq) collection (https://ftp.ncbi.nlm.nih.gov/genomes/refseq/), was used as the reference database. The result were the raw abundance profiles of prokaryotes (bacteria, archaea), eukaryotes (protozoa, metazoa etc.,) and viruses, stratified across all taxonomic levels.

Downstream metagenomic analysis

Data filtering and data normalization steps were performed to remove low quality or uninformative features from raw abundance data to improve downstream statistical analysis. Briefly, features with exceedingly small counts (<5 reads) and in very few samples (<10% prevalence) were filtered out, followed by a low variance filter using variances measured by inter-quantile range (IQR). Normalization is an essential step in the analysis of microbial abundances in shotgun metagenomics. Data normalization addressed the variability in sampling depth and any sparsity of the data to enable more biologically meaningful comparisons. Trimmed mean of M-values (TMM) is one of the best performing normalization methods, which has showed a high True Positive Rate (TPR) and low False Positive Rate (FPR) (Pereira et al., 2018). It is also known to be best in controlling the FDR. Hence, we performed TMM normalization on the data to ensure accurate biological interpretation of the metagenomic data.

Taxonomic composition of communities across samples and comparing groups were visualized for direct quantitative comparison of abundances. Percentage bar plots were created for comparing group of the test arm, Day 1 (before intervention) and Day 90 (after intervention), for viewing the composition at various taxonomic levels.

Alpha diversity was characterized using different measures. Chao1 index was used for richness-based measure, while Shannon index was used to estimate diversity of the community based on richness as well as evenness (the abundance of organisms). Further, the statistical significance of grouping based on experimental factor was also estimated. Furthermore, ‘similarity’ or ‘dissimilarity’ between the two experimental factors was also measured using Beta diversity methods. Non-phylogenetic beta diversity analysis was performed employing Bray-Curtis distance. Principle coordinate analysis (PCoA) was used to visualize the distance matrix created by the beta diversity analysis and statistical significance of the clustering pattern in PCoA plots were evaluated using Permutational ANOVA (PERMANOVA). Both the alpha and beta diversity analyses were performed using the phyloseq packages (McMurdie & Holmes, 2013, 2015) (https://github.com/joey711/phyloseq) and the results were plotted as box and whisker plots for alpha diversity and PCoA plot for beta diversity respectively.

Differential abundance (DA) analysis was also performed to identify and characterize significantly altered microbial abundances across the experiment factors. Recently it has been highlighted that there is a high variation in the output of DA tools across sequencing datasets, presenting issues with reproducibility among microbiome researchers. Hence, it is recommended that researchers use a consensus approach based on several DA tools to help ensure results are robust (Nearing et al., 2022). Considering this, we performed the differential abundance analysis with five different DA tools, viz., Univariate Analysis (T-Test ANOVA) (Luz Calle, 2019), MetagenomeSeq (Paulson, 2016; Paulson, Talukder & Bravo, 2017; Paulson et al., 2013), EdgeR (v3.12) (Robinson, McCarthy & Smyth, 2009), DeSeq2 (Love, Huber & Anders, 2014), and LEfSe (Linear discriminant analysis Effect Size) (Segata et al., 2011). While each of these DA tools differ in their approach to data normalization and the algorithms used for evaluation of variance or dispersion, features were deemed to be significant based on their adjusted p-value (default adj.p-value cutoff = 0.05). Once the DA analysis was performed using individual tools, we identified those microbial species that were called significantly (p < 0.01) differentially abundant in “consensus” by three or more DA tools, ensuring the robustness of the DA characterization.

In order to gain insights into the probable role of taxa in terms of correlation and deduce the importance of their participation in biological interactions, we also performed the network analysis. In order to illustrate the differential correlations of the gut metagenome profiles before and after the BugSpeaks® personalized diet, analysis was conducted for selected taxa obtained after data pre-processing and only those significantly correlated taxa were reported. Briefly, abundance profiles across all samples were imported and loaded using Pandas (V. 2.1.2, https://pandas.pydata.org/), and preprocessing of the data was performed, which included removal of genus and species with low variance and low raw abundance counts. TMM transformation of the data was performed using Conorm (V. 1.2.0), and other pre-processing steps were performed using Sklearn (V. 1.3.0), which together created train test splits, for using this data to train. Spearman R correlation, between all species/genus and the intervention arm, was performed using Scipy (V. 1.11.1), followed by a filtration step of removing all the correlations with <0.5 Spearman R coefficient and >0.05 p-value. Network diagrams of species interactions of pre- and post-intervention groups were also generated using Networkx (V. 3.1).

Efficacy and safety variables

All endpoints were set to assess impact of gut microbiome-based dietary intervention. It included the estimation change in serum HbA1c, CRP, total cholesterol, LDL, HDL, triglycerides, non-HDL cholesterol and IL-10, and change in faecal gut microbiome. Additionally, assessment of adverse events, vital signs (pulse rate, systolic and diastolic blood pressure (seated), BMI, body temperature and respiratory rate and physical examination was done to evaluate safety of the intervention.

Data availability

The datasets generated from the next-generation sequencing in this study is available in the NCBI Sequence Read Archive (SRA) repository, Bioproject ID: PRJNA1046298.

Results and Discussion

Present clinical trial was conducted to evaluate the safety and impact of BugSpeaks® microbiome-based personalized dietary regime in hyperglycaemic and hyperlipidaemic individuals, specifically to evaluate the impact of such diet on gut microbiota and other disease related clinical parameters. Please note, that portions of this text were previously published as part of a preprint (Kallapura et al., 2023). The current clinical trial was a randomized, double-blinded, prospective study, initiated with 30 Indian subjects with hyperglycaemia and/or hyperlipidaemia per study design (Fig. 1, study flow chart). Demographic details of the test subjects at baseline were provided in Table 1 (subject characteristics at baseline). The effect of the intervention on HbA1c, total cholesterol, LDL, HDL, triglycerides, non-HDL cholesterol, CRP and IL-10 levels were determined on 90th day of the study, post 3 months of dietary intervention. The changes in gut microbiome profiles were characterized for the test arm only, before and after the microbiome-based intervention. Further, safety of the gut microbiome-based personalized diet was studied in terms of the adverse events.

Table 1 Table-subject characteristics at baseline.

Parameters	Male	Female	
n	19	11	
Age (Years)	55.05 ± 7.0	53.82 ± 7.97	
Height (cm)	170.42 ± 6.79	165.45 ± 7.19	
Weight (kg)	72.63 ± 11.37	66.73 ± 10.69	
BMI	24.95 ± 3.11	24.32 ± 2.89	
Note:

Values represented as mean ± standard deviation.

Significant reduction in HbA1c levels

Statistically significant decrease in HbA1c level was observed in the test arm with personalized microbiome-based diet from 8.30 (95% CI [7.74–8.85]) to 6.67 (95% CI [6.2–7.05]), p < 0.001, while only small numerical non-statistical decrease in HbA1c level was observed in the control arm with regular diet from 8.24 (95% CI [7.7–8.6]) to 7.32 (95% CI [6.22–8.4]), p = 0.15, after 90 days of dietary intervention (Fig. 2A). A total of 100% of the participants in the test arm, who followed microbiota based personalized diet, showed a decrease in HbA1c levels, with a mean reduction of 1.62% in HbA1c absolute count (Fig. 2B), while only 78.5% participants in the control arm showed a decrease in HbA1c levels, with a mean reduction of only 0.91% in HbA1c absolute count (Fig. 2C). This meant that there was a 19.6% drop in mean HbA1c levels in the test arm with microbiome-based dietary intervention, as compared to only a 11.1% drop in the control arm (Fig. 2D). This strongly indicated that achieving a significant reduction in HbA1c levels is possible with personalization of diet based on one’s gut microbiome. The reduction in HbA1c was found to be much more profound as compared to diabetic specific diet in this trial (test arm vs control arm). This reduction in HbA1c levels was also correlated with changes in the gut microbiota, with shift in composition, abundance and diversity of several species within the gut (detailed below).

Figure 2 Change in HbA1c levels.

(A) Overall change in HbA1c levels across the arms, where ***p < 0.001. (B and C) Change in HbA1c levels in each individual, within the BugSpeaks personalized nutrition arm and the regular nutrition arm, respectively. (D) Overall % drop between the two arms.

Significant decrease in systolic pressure

Elevated blood pressure is a major cardiovascular and metabolic disease risk factor (Valles-colomer et al., 2023). Gut microbiota dysbiosis has been reported in patients with high blood pressure (Dan et al., 2019). Gut microbiota modulation has been shown to impact blood pressure (Yan et al., 2022). Hence, we wanted to investigate if microbiota based nutritional intervention would impact the blood pressure parameters in the participants of the test arm. Mean systolic blood pressure was found to be slightly reduced in the participants of the test arm 139 mm Hg (95% CI [127.8–150.1]) to 132 mm Hg (95% CI [126.4–137.5]) post intervention whereas it was found to slightly high in the participants of the control arm who undertook regular nutrition 126 mm Hg (95% CI [122.5–129.4]) to 135 mm Hg (95% CI [127.9–142.1]). This change however was not statistically significant (Fig. 3A). Interestingly, statistically significant decrease in systolic pressure 153 mm Hg (95% CI [138.7–167.2]) to 131 mm Hg (95% CI [125.8–137.7]), p < 0.01 was found in a subset of the participants (eight out of 15) in the test arm at the end of the study period (day 90) whose basal systolic pressure was >130 mm Hg prior to the microbiome based personalized dietary intervention. Similarly, a 4.5% decline in diastolic pressure was also found in this subset of participants from the test arm, although this decrease was not statistically significant (Fig. 3B). It has been reported that increase in Lactobacillus and Bifidobacteria are associated with lower blood pressure (Yan et al., 2022). Interestingly, gut microbiota analysis of these participants showed increased abundance of phylum Firmicutes (Lactobacillus is member of this phylum) and Actinobacteria (Bifidobacteria is a member of this phylum) (Fig. 4A). More specifically Levilactobacillus brevis and Bifidobacterium angulatum were found higher post intervention in the test arm participants. At the genus level we observed an increased abundance of Roseburia and Bacteroides, and decreased abundance of Prevotella and Phocaeicola post intervention with personalized diet (Fig. 4B). Also, our analysis showed a decreased abundance of Alistipes finegoldii in the participants post intervention. Strikingly, high abundance of Alistipes finegoldii has been reported in the intestine of patients with high blood pressure (Kim et al., 2018). So, a reduction in Alistipes finegoldii abundance might also be a contributing factor in the improvement of the blood pressure parameters in this group.

Figure 3 Change in blood pressure parameters.

(A) Overall change in systolic and diastolic pressures across the comparing arms of regular nutrition and BugSpeaks nutrition. (B) Change in systolic and diastolic pressures within a sub-cohort of patients in BugSpeaks nutrition showing a significant reduction in systole within the arm, with **p < 0.01.

Figure 4 Differential abundance across phylum (A) and genus (B) levels.

Lower serum CRP levels

Chronic inflammation has been found to be associated with type 2 diabetes, hyperlipidaemia etc., (Han & Lin, 2014; Guo et al., 2023). High CRP level is an indicator of inflammation and underlying disease conditions such as cardiovascular diseases and type 2 diabetes (Bafei et al., 2023; Kuppa et al., 2023; Guo et al., 2023). We evaluated the change in CRP level in a subset of participants whose basal serum CRP level was >= 2 mg/L prior to intervention (nine out of 15 participants). We found 20% decrease in the CRP level post intervention although the decrease was not statistically significant (Fig. 5). Interestingly, we found decreased Prevotella and increased Roseburia along with an increased Levilactobacillus brevis in these participants post intervention (Fig. 4B). This might be one of the reasons for reduced inflammation as increased Prevotella has been found to have pro-inflammatory effect (Larsen, 2017). Similarly increased Levilactobacillus brevis and Roseburia have been associated with reduced inflammation (Nie et al., 2021; Riccia et al., 2007; Fernández-Veledo & Vendrell, 2019; Riccia et al., 2007; Gupta et al., 2020; Wei et al., 2023). Lowered CRP levels after dietary intervention further shows the positive impact of the microbiome-based personalized nutrition in chronic disease conditions such as Type 2 Diabetes. Increase in subject population, along with further customization of the microbiome-based diet, might be helpful to get statistically significant changes in CRP levels.

Figure 5 Change in serum CRP levels.

CRP levels were found to be decreased by 20% post intervention by BugSpeaks® personalized nutrition.

Change in other endpoints

No statistically significant decrease in the levels of total cholesterol, LDL, HDL, triglycerides, non-HDL cholesterol and IL-10 was observed in the test arm with personalized microbiome-based diet, as compared to the control arm with regular diet, after 90 days of intervention.

Significant changes in gut microbiome diversity and species abundances

Change in gut microbiome profiles after 90 days of intervention with microbiome-based personalized dietary regime was characterized only for the test arm and was visualized for direct quantitative comparison of abundances, followed by alpha and beta diversity measures, and lastly differentially abundant species and network and correlation analysis across the comparing groups.

Since we performed the whole metagenomic sequencing, we were able to profile all the microbial taxa within the sample, including fungi and viruses. We did not observe any significant difference in the composition; however, we did observe some shift in abundance and diversity of few groups. Largely, abundances of Bacteria, Archaea, and Viruses were slightly decreased by Day 90 of microbiome-based intervention, by 0.07% (from 99.44% to 99.37%), 0.04% (from 0.12% to 0.08%), 0.03% (from 0.13% to 0.10%) respectively. Inversely, abundances of Fungi, and Eukaryota increased slightly by Day 90 of microbiome-based intervention, by 0.08% (from 0.20% to 0.28%) and 0.06% (from 0.11% to 0.17%) respectively. In context of diversity, Shannon diversity index estimated an increase in diversity in Bacteria, while small decrease in diversity in kingdoms Archaea, Fungi, Eukaryota and Virus (Fig. 6). Other patterns emerged at phylum level, with a significant decrease in abundance of Bacteroidetes (from 76.10% to 66.19%), with a 9.91% shift, post microbiome-based intervention. This reduction in abundance of Bacteroidetes was largely attributed to the net shift in abundance of genus Prevotella, with decrease in abundance of Prevotella copri (↓ by 9.79%), Phocaeicola plebeius (↓ by 6.54%) and Prevotella hominis (↓ by 1.56%), and increase in abundance of Pseudonocardia cytotoxica (↑ by 1.44%), Prevotella stercorea (↑ by 1.51%) and Bacteroides_sp_CBA7301 (↑ by 4.45%). There was substantial increase in the abundances of Firmicutes and Actinobacteria in test arm, with 5.57% increase (from 16.09% to 21.66%) and 2.84% increase (from 0.98% to 3.82%), respectively. Many butyrate producing bacteria and probiotics are from these phyla and hence it is possible that a positive shift in these phyla may have led to an improvement in hyperglycaemic and inflammation parameters in this current study.

Figure 6 Changes in diversity of gut microbiome, within the BugSpeaks nutrition arm.

Changes in Shannon alpha diversity measure across kingdoms

Alpha diversity measures further confirmed these abundance shifts, with significant increase in Shannon diversity measure, from 2.43 to 3.11 (p = 0.029), post microbiome-based personalized dietary intervention (Fig. 7A). On the other hand, Chao1 indicated a minor decrease in diversity, from 1,154 to 1,126 species (Fig. 7B). Together, these estimates indicated that there was an overall decrease in species richness, while a significant increase in species evenness. In other words, the microbiome-based dietary intervention impacted the gut microbiota by reducing the number of species by a small degree, while modulating the other abundances of other species, overall displaying evenness in context of diversity. Similar observation was found by Gupta et al. (2020) in their study that showed that subjects in the healthy cohort had higher Shannon diversity whereas species richness was higher in the subjects of the “unhealthy cohort”. So, it can be speculated that the lower species richness and higher Shannon diversity in the intervention arm is possibly better in the context of health of the subjects. The beta diversity measure by Bray-Curtis distance didn’t establish any significant difference between the two arms. However, it displayed clustering between the groups, represented by the ellipses in Fig. 7C. Increase in subject population, along with finetuning of the personalization of the microbiome-based diet, might aid in statistically significant changes in diversity measures.

Figure 7 Changes in Shannon and Chao1 diversity of gut microbiome.

(A) Significant change in Shannon alpha diversity measure post intervention with BugSpeaks Nutrition. (B) Changes in Chao1 diversity measure between pre and post BugSpeaks nutritional intervention. (C) Beta diversity measure by Bray-Curtis distance between pre and post BugSpeaks nutritional intervention.

We observed changes in microbiome profile at the higher taxonomic levels (Phyla), specifically Firmicutes and Actinobateria phyla showed increase whereas Bacteroidetes phyla showed decrease in the participants post BugSpeaks® intervention (Fig. 4). We also observed several specific changes at species level that were statistically and potentially functionally significant as highlighted below. Based on the consensus approach employed with five different differential abundance (DA) tools, we could establish as many as 15 species to be significantly differentially abundant (p < 0.05) between the two arms of the study. We estimated a minimum 2-fold (Log2 fold change) increase in Brachyspira pilosicoli, Phascolarctobacterium succinatutens, Phascolarctobacterium sp Marseille Q4147, Bifidobacterium angulatum, Acinetobacter venetianus, Levilactobacillus brevis, and Acidithiobacillus ferriphilus in the test arm. On the other hand, we estimated a minimum 2-fold (Log2 fold change) decrease in Parabacteroides sp ZJ 118, Sutterella seckii, Sutterella sp KLE1602, Phocaeicola sp Sa1CVN1, Alistipes finegoldii, Sutterella faecalis, Lachnoclostridium pacaense, and Treponema succinifaciens, in the test arm with microbiome-based personalized dietary intervention. The comparative abundance plots of some of these species are displayed in (Figs. 8A–8H)

Figure 8 Significantly differentially abundance species.

(A) Log2 Fold Change, of some of the most differentially abundant species; (A–H) Differential abundance of species that were differentially abundant (*p-value < 0.05).

Some of the more interesting correlations, between the above highlighted reduction in HbA1c and CRP levels, were observed at species level. To begin with, maintenance of optimal levels of succinate is key during glucolipid metabolism, where succinate regulates glucose homeostasis to ameliorate hyperglycemia (Wei et al., 2023).

Phascolarctobacterium succinatutens belonging to the Negativicutes class of Firmicutes was found to be significantly higher in abundance within the test arm post microbiota based nutritional intervention. P. succinatutens is a species of asaccharolytic (does not ferment sugars) bacteria, has been previously isolated and identified from the healthy human gut, known to play a key role in governing intestinal homeostasis and energy metabolism (Watanabe, Nagai & Morotomi, 2012; Ikeyama et al., 2020). The most important characteristic of P. succinatutens is that it is a succinate-utilizing bacterium, that exclusively uses succinate produced by other bacteria (such as Bacteroides species) as the substrate for propionate production (Watanabe, Nagai & Morotomi, 2012; Sawaswong et al., 2023; Fernández-Veledo & Vendrell, 2019; Muhammad et al., 2023). Further, a Mediterranean (plat rich) diet has been found to increase the ratio of succinate-consuming bacteria (Like, P. succinatutens, Odoribacteraceae and Clostridaceae) to succinate producing bacteria (like Prevotella copri, and other species of Prevotellaceae and Veillonellaceae) (Wei et al., 2023; Sawaswong et al., 2023). This pattern was also observed in the current study, where the succinate producing Prevotella copri was observed to be reduced in abundance by 9.791%, while the succinate consuming P. succinatutens was found to be significantly increased in its abundance (by 4.5-fold (log2 fold change)), post implementation of microbiome-based personalized diet. Further, as highlighted above, maintenance of optimal levels of succinate is key during glucolipid metabolism to regulate glucose homeostasis and ameliorate hyperglycemia (Wei et al., 2023). It would be interesting to recommend a larger study with simultaneous measurements of HbA1c, succinate and other serum parameters to confirm this observation. This may open up prospects for using specific succinate consuming bacteria that are beneficial to host health or to administer succinate-consuming probiotics and promote their growth through high fibre dietary intervention, which is expected to lead to the uptake of excessive succinate and provide new avenues for treating related diseases (Muhammad et al., 2023; Sharma, Bhardwaj & Singh, 2016; Bernier et al., 2021; Chen et al., 2023).

Decreasing abundance of Prevotella and increasing abundance of Phascolarctobacterium along with Levilactobacillus brevis and Roseburia could be the correlating factor for the observed reduction in inflammation. Furthermore, succinate has potential as a target for immune monitoring (Wei et al., 2023; Macias-Ceja et al., 2019), and recently, reducing the succinate concentration has shown promise in treating gut chronic inflammatory diseases and obesity-related inflammation, suggesting a new way to alleviate these diseases (Serena et al., 2018; Fremder et al., 2021). Hence, the personalization of diet based on one’s gut microbiome might have true potential in addressing various inflammatory diseases.

Few more species of bacteria, with potential probiotic properties were also found to be significantly increased in the test arm who adopted the microbiome-based personalized diet. Bifidobacterium angulatum and Levilactobacillus brevis, both were estimated to be increased in the test arm by 3.485- and 2.213-fold (Log2 Fold Change), after 90 days of personalized diet regime. Bifidobacterium angulatum is a species of bacteria that is part of the human gut microbiota, a relatively less common species Bifidobacterium group of probiotics (Zakharevich et al., 2019). Administration of other Bifidobacterium probiotics, such as Bifidobacterium bifidum and Bifidobacterium breve have been associated with amelioration of hyperglycaemia, dyslipidaemia, and oxidative stress in various studies (Fu et al., 2022; Sharma, Bhardwaj & Singh, 2016; Bernier et al., 2021). The observation of this study also indicates the potential of Bifidobacterium angulatum in amelioration of hyperglycaemia and reduction of HbA1c levels. On the other hand, Levilactobacillus brevis has been previously reported to alleviate the progression of type 2 diabetes in animal models, via interplay of gut microflora, bile acid and NOTCH 1 signalling (Chen et al., 2023). Also, Levilactobacillus brevis possess inhibitory effects on α-amylase and α-glucosidase activities, and has been reported to have anti-diabetic properties (Martiz et al., 2023). Hence, the significantly reduced levels of HbA1c in this study, might be a direct correlation to the increased abundance of Levilactobacillus brevis.

Network diagrams of species interactions of pre- and post-intervention, with all statistically significant associations (Spearman coefficient >0.5 and p-value < 0.05), have been shown in Figs. 9A and 9B. Within these network analysis comparisons, few key negative correlations were observed between species, especially between Sutterrella sp KLE1602 and Phocaeicola massiliensis (Spearman correlation coefficient −0.66) (Fig. 10C) in the pre-intervention group of the test arm. Interestingly, Phocaeicola massiliensis is observed in a positive correlation with Parabacteroides distasonis both in the pre and post intervention participants (Figs. 9A and 9B). Recent report suggests that higher abundance of Parabacteroides distasonis may alleviate metabolic syndrome through production of succinate (Wang et al., 2019). As we find that Sutterrella sp KLE1602 is reduced in post intervention group, it is tempting to speculate that in the pre intervention group due to higher abundance of Sutterrella sp KLE1602, it may indirectly reduce Parabacteroides distasonis level thereby causing some of the metabolic syndrome effects. Clustering based on Spearman correlation shows an interesting pattern wherein some clusters of microbes are seen to be positively correlated among each other in the post intervention as compared to the pre intervention group (Figs. 11A and 11B). This is an interesting pattern and we do not know the reason or the possible implication. We can speculate that this is due to the microbiota modulation through individualized diet however, it would be interesting to find out if this shift in the correlation pattern influences the clinical outcome in the participants of the test arm in subsequent trials.

Figure 9 Network analysis.

(A) Network of associations pre-intervention, along with (B) network of associations post-intervention with BugSpeaks personalized nutrition. The features of the network include: edge thickness is proportional to the Spearman correlation between species, edges are deleted if Spearman is below the thresholds highlighted above, blue-coloured edge represents a negative correlation, while a pink-coloured edge represents positive correlation; finally different node colours were used for different kingdoms.

Figure 10 (A–C) Correlation analysis.

(A and B) Specifically showing positive (high Spearman correlation coefficient) and negative (low Spearman correlation coefficient) correlations between two species.

Figure 11 Hierarchical clustering pre and post intervention.

Heat maps representing the clustering of species based on Spearman correlation. (A) Pre-intervention and (B) post-intervention. Higher number of positive correlations were observed post intervention with BugSpeaks® personalized nutrition.

Conclusion

Gut microbiota has been implicated in various chronic diseases such as type 2 diabetes (Martínez-López et al., 2022; Cunningham, Stephens & Harris, 2021). Modulating gut microbiota through dietary interventions such as prebiotics and probiotics have shown promising results in improving disease conditions (Campaniello et al., 2023; He & Shi, 2017). Propositions have also been made to modulate diet to improve the efficacy of treatment of COVID-19, anxiety and depression (Dhar & Mohanty, 2020; Dhar, 2022). In this study we evaluated the efficacy and safety of a personalized gut microbiota specific nutritional intervention utilizing next generation sequencing based profiling of the individual gut microbiota. Our study shows marked improvement in the hyperglycaemic, hypertensive and inflammation parameters in the participants of the test arm who followed diet plan based on their unique gut microbiota profile as compared to the control arm participants who followed diabetic specific regular nutritional plan. The small decrease in the HbA1c level in the control arm can be attributed to the fact that the participants of both the test and control arms continued with their stable dose of anti-diabetic medication during the trial as was ethically required for the study. The participants in the control arm also received a dietician and Principal Investigator approved diabetic meal plan and hence the possibility of the diabetic regular meal plan’s effect on lowering the HbA1c level in this arm can also be one of the contributing factors. The study shows that gut microbiota directed diet plans can supplement the anti-diabetic medication and improve upon the overall condition in such patients. We observed a significant decrease in hypertension parameters in the participants of the test arm thereby highlighting the overall positive impact of the personalized gut microbiota-based diet intervention. However, we did not find any significant decrease in hyperlipidaemic parameters such as total cholesterol, HDL and LDL levels in this study. This might be because of lesser duration of the study to see the impact of microbiota based personalized nutrition on such parameters. Overall, no adverse effect was reported by the participants following the gut microbiota based nutritional meal plan thereby showing the safety of such an intervention. As we provided gut microbiota based nutritional recommendation to the participants of the test arm, we expected a modulation of their gut microbiota and hence their microbiota were profiled post intervention. We did not profile the gut microbiota of the control arm post routine nutritional intervention however we do not rule out subtle microbiota changes in the participants of that cohort as well.

Our study also found specific changes in the gut microbiota post intervention that may have contributed to the positive effects. Specifically, we found increased Shannon diversity post intervention and possibility of better utilization of succinate in that group. Increased abundance of bacteria such as Bifidobacterium angulatum and Levilactobacillus brevis may have also contributed in improving hyperglycaemic, hypertensive and inflammation parameters (Fig. 12). Many bacteria belonging to the genera Bifidobactria and Levilactobacilli are known probiotics. This trial to our knowledge, is the first of its kind study in Indian patients thus emphasising the positive impact of gut microbiota modulation in disease irrespective of the ethnicity.

Figure 12 Trial summary.

The trial showed that BugSpeaks®personalized nutrition led to improvement in HbA1c, CRP and blood pressure parameters in Type 2 diabetic patients. This improvement may be attributed to an increase in beneficial microbes such as Bifidobacterim angulatum and Levilactobacillus brevis. Possible mechanism may also include better balance between succinate producers and consumers in the host leading to appropriate concentration of succinate in the system. All icons, graphics are used by utilizing Canva pro account (www.canva.com).

In totality, personalized nutrition based on one’s gut microbiome aims to preserve or increase the overall gut health using relevant information about the individual’s gut microbiome, by delivering personalized nutritional recommendations (Vandeputte, 2020). Such personalization of nutritional advice will be far more effective than more generic approaches and future of personalized nutrition strategies would rely significantly on the gut microbiome to manage disease conditions and overall health (Vandeputte, 2020; Lee, Davies & Barnett, 2023; Bianchetti et al., 2023; Aarnoutse et al., 2017). While the potential of such an approach is promising, it’s important to note that our understanding of the gut microbiome and its complex interactions with our bodies and our diets is still evolving. More research is needed to fully understand the potential benefits and challenges of a microbiome-based personalized diet. Personalized nutrition based on one’s gut microbiome offers a promising approach to rectify dysbiosis and improve health outcomes (Song & Shin, 2022; Hernández-Calderón, Wiedemann & Benítez-Páez, 2022; Vandeputte, 2020).

Supplemental Information

Supplemental Information 1 Normalized percentage abundance across groups.

Supplemental Information 2 Raw Data - Hypertension parameters.

Supplemental Information 3 Raw Data- HbA1c.

Supplemental Information 4 Raw data - CRP.

Supplemental Information 5 CONSORT checklist.

We acknowledge Deeksha Garg for helping with the generation of figures.

Additional Information and Declarations

Competing Interests

Author Contributions

Human Ethics

Clinical Trial Ethics

Data Availability

Clinical Trial Registration

Gopalakrishna Kallapura, Anthony Surya Prakash, Kumar Sankaran, Prabhath Manjappa and Debojyoti Dhar are employed by Leucine Rich Bio Pvt Ltd and Sanjay Ambhore by Shreya Clinic.

Gopalakrishna Kallapura conceived and designed the experiments, performed the experiments, analyzed the data, prepared figures and/or tables, authored or reviewed drafts of the article, and approved the final draft.

Anthony Surya Prakash performed the experiments, authored or reviewed drafts of the article, and approved the final draft.

Kumar Sankaran conceived and designed the experiments, authored or reviewed drafts of the article, and approved the final draft.

Prabhath Manjappa conceived and designed the experiments, authored or reviewed drafts of the article, and approved the final draft.

Prayagraj Chaudhary analyzed the data, prepared figures and/or tables, and approved the final draft.

Sanjay Ambhore performed the experiments, authored or reviewed drafts of the article, and approved the final draft.

Debojyoti Dhar conceived and designed the experiments, analyzed the data, prepared figures and/or tables, authored or reviewed drafts of the article, and approved the final draft.

The following information was supplied relating to ethical approvals (i.e., approving body and any reference numbers):

The Institutional Ethics committee of Charak Hospital-Reg: ECR/152/Inst/MP/2021, Bhopal, India granted Ethical Approval to carry out the study (Approval ref: PRO2022/03/03).

The following information was supplied relating to ethical approvals (i.e., approving body and any reference numbers):

Institutional Ethics committee, Charak Hospital-Reg: ECR/152/Inst/MP/2021, Bhopal, India.

The following information was supplied regarding data availability:

The datasets generated from the next-generation sequencing in this study is available at NCBI Sequence Read Archive: PRJNA1046298.

The following information was supplied regarding Clinical Trial registration:

ctri/2022/05/042791.

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
