# Peer review of "Microbiota based personalized nutrition improves hyperglycaemia and hypertension parameters and reduces inflammation: a prospective, open label, controlled, randomized, comparative, proof of concept study"

_PeerJ, doi:10.7717/peerj.17583_

## Round 0.1 · original submission · Major Revisions

Dear Dr. Dhar,

Your manuscript entitled “Microbiota based personalized nutrition improves hyperglycaemia and hypertension parameters and reduces inflammation: a prospective, open label, controlled, randomized, comparative, proof of concept study" which you submitted to PeerJ, has been reviewed by the editor and 2 external reviewers.

I regret to inform you that the reviewers have raised some significant concerns that need to be addressed before the manuscript can be considered further. However, since reviewers do find some merit in the paper, I would be willing to reconsider if you wish to undertake major revisions and resubmit.

If you decide to resubmit the revised version, please summarize all the improvements made in the new version and give answers to all critical points raised in the reviewers’ report in an accompanying letter. Copy and paste each and every reviewer's comment above your response.

Please note that resubmitting your manuscript does not guarantee eventual acceptance. Since the requested changes are major, the revised manuscript will undergo a second round of review by the same reviewers. I must emphasize that the acceptability of the revision will depend upon the resolution of the points raised by the reviewers.

Sincerely yours,
Stefano Menini

·

Basic reporting

View below

Experimental design

View below

Validity of the findings

View below

Additional comments

The manuscript presents a prospective interventional trial investigating the impact of BugSpeaks® recommended gut microbiome-based personalized diet in hyperglycemic and hyperlipidemic individuals. Below, I have provided detailed feedback on various aspects of the manuscript based on the notes provided.

Line 62: There is a missing dot after "insulin resistance" in the phrase "branched-chain amino acids causing insulin resistance 3,2,9,10 For instance,".

Line 88: The description of BugSpeaks® lacks coherence with the preceding context. Specifically, the transition from discussing BugSpeaks® to its integration into the precision medicine paradigm lacks clarity.

Line 92: The statement regarding the evaluation of clinical parameters lacks reference support to underscore their significance. Consider referencing the importance of these parameters with the following sources: https://doi.org/10.1111/eci.13455, https://doi.org/10.1007%2Fs13300-016-0167-x, and https://doi.org/10.1016/j.neurobiolaging.2015.10.039.

Line 110: “Both male and female participants were included in the study, with an age range of 42-65 years; with varying body weights [body mass index [BMI] of 19.6 3 33.3 kg/m2];” Please provide mean and standard deviation for these values.

Line 127: Avoid the repetitions of “After” in this text: “After obtaining the informed consent, subjects were screened by undergoing various assessments. After confirming eligibility, eligible subjects were randomized to receive either”.

Line 130: “Subjects in both the arms continued with their stable dose of diabetic medication. (Sulphonylureas,DDP4 Inhibitors,Thiazolidinediones,Nateglinide)” Please move the point after the parentheses. And change “in both the arms” to “in both arms”.

Line 133: “daily patient diary recording.” How was the data recorded for the daily patient diary? Additionally, recent studies have underscored the impact of food monitoring, such as self-monitoring, on metabolism and weight loss [https://doi.org/10.3390/jpm12040568, https://doi.org/10.3390/su15010614]. Has this factor been considered in your study?

Line 135: “2.2 Investigation product [IP]: BugSpeaks®” How are these diets formulated? It is crucial to provide a detailed scheme to clarify the nature of the diet. While the description states, “Personalized diet plans based on the individual’s microbiota profile were generated using algorithms and matrices which took into consideration abundance of various microbes and the effect of various food items in modulating their level” there is insufficient information to fully understand the process. Please provide a more comprehensive description.

Line 149: “[www.randomization.com]” Please also provide the last access date.

Line 169: “are mentioned below:” Please remove the highlighting in “:”.

Line 170: This appears to be a table; however, no caption or description has been provided for context or clarity.

Line 171: Have you assessed the suitability of the t-test conditions, such as checking for normality of distributions? Additionally, could you provide details on how these analyses were conducted and specify the software utilized?

Line 278: “Pandas [V. 2.1.2],” Please also provide the homepage of the library.

Line 286: “The features of the network include;” Change “;” in “:”.

Line 301:“(https://www.ncbi.nlm.nih.gov/bioproject/PRJNA1046298).” The format of this URL differs from others. Additionally, please maintain consistency in reporting URL formats. Replace "(" with "[", and ensure to include the last access date.

Line 334: “disease risk factor44 .” Remove the space after the citation.

Line 342: “confidence interval (CI), 122.5-129.4] to 135mm Hg [95% confidence interval (CI), 127.9-142.1] .” Remove the space after the “]”.

Line 365: Change "type 2 diabetes" to "Type 2 Diabetes" to maintain consistency with the formatting used throughout the manuscript.

Line 398: “[from 76.099% to 66.186%],” Ensure consistency in the level of approximation by changing to 76.10% and 66.19%.

Line 383: “Change in gut microbiome profiles after 90 days of intervention with microbiome-based personalized dietary regime was characterized only for the test arm and was visualized for direct quantitative comparison of abundances,”.

Why did the comparative diet fail to impact the microbiome? This warrants discussion and a clearer explanation of the diet type and its purpose. The term "diabetic meal plan" was used, but lacks elaboration. Please provide a detailed explanation in the methods section.

Line 421: “Even though there were no statistically significant changes in microbiome profile at the higher taxonomic levels, except with Bacteroidetes,”

How is it plausible that a microbiome-tailored diet yielded no discernible impact on the microbiome? Clarify the nature of the diet and its intended objectives in the methods section. Additionally, elucidate whether the study found any unexpected outcomes and provide explanations for any discrepancies between expected and observed results.

Figure 5: What is the rationale for utilizing a 3D bar plot? If the third dimension is unnecessary, consider using a 2D bar plot instead for clearer visualization and interpretation.

Figure 6: The labels on the figures are too small, making them difficult to read. Please resize them to ensure readability, with a minimum font size of 10. This adjustment should be applied to all figures.

Figure 8: The labels in panels A and D are not legible. Please resize them to a minimum font size of 10 to improve readability.

Figure 9: Please resize the figure or consider dividing it into two separate parts. The labels are currently unreadable. Alternatively, if the labels are unnecessary, they can be omitted, and their meaning can be explained in the figure's description.

Table 1: Revise "Hight (cm)" to "Height (cm)" for consistency. Remove the "Gender" column to maintain uniformity in the table structure. Replace it with "Parameters" as the header instead. Identify "Male" and "Female" as the respective groups within the table.


Overall, the manuscript has potential, but revisions are necessary to address the outlined issues and improve the clarity and coherence of the content. Once the revisions are made, the manuscript would be suitable for further consideration.

·

Basic reporting

A few ambiguous statements and/or overstatements are found throughout the text. Namely: - within the abstract the authors mention ‘beneficial species’ and ‘non-beneficial species’, this seems to much of a simplification. It might be that these species are pursued beneficial in the context of the current pathology, but they might not have this connotation in every context. I would therefore advice a more careful wording.
- Lines 66 to 68 states the gut microbiota influences host nutritional status and phenotype, regulating everything from nutrient absorption, storage, metabolism to even disease development and progression. This is too strongly worded. Most likely the authors meant that the gut microbiota has an impact on a variety of processes implicated in metabolic syndrome (not syndromes, btw) and these sentences should be rephrased accordingly.
- Line 71-72: malleability of the gut microbiome is a weird expression. But more importantly, I don’t think its correct to state that gut microbial composition can be changed ‘easily’ through the diet, though easy is of course a subjective term. Microbial abundances can indeed shift rapidly upon dietary changes, but which bacteria are present or absent, is much more stable. I would advise on revising the text so that it reflects these insights.
- Line 416: “modulating the other species to an even ‘and better’ distribution of abudances.” Is an even distribution ‘better’? Why would that be? And would that be the case in all circumstances?
- Line 527, it is confusing to call bacteria identified within this study ‘probiotics’, even though this study (and others) indicate they have beneficial effects on health, while they were not administered to the patients. They might belong to genera that include well-known probiotics, but they were not used as such in this study. Using the term as an adjective is confusing in this context.
It is unclear how BugSpeaks works, how it comes to its recommendations, if these are reproducible, and what they were.
Lines 286-289 should be part of the figure legend and not the method section.
The result and discussion section would benefit from more informative subtitles that convey the message of the paragraph.
It is hard to know whether the description about the significant genera covers all of those that were found significantly altered or only highlights a few interesting ones. A complete overview of all significant genera, possibly including an indication of previous findings, in table format, would be a nice addition to the manuscript.
Recommendations for further work are sometimes quite abruptly stated. These sentences might be improved by some introductory part identifying it as ‘recommendation for further work’. E.g. line 454.
Please update the figures so that they align with each other. Specifically, convert 3D figures to 2D figures. The 3D does not add anything to their understanding. Please indicate significance (or absence thereof) between values within the figure where they are currently lacking (e.g. Fig 5; Fig 6)

Experimental design

According to the exclusion criteria, antibiotic use in the two weeks prior to the study would lead to exclusion. This is a quite short time frame given that effects of antibiotics are observed for much longer timeframes. Where there any participants that took antibiotics in the last 6 months and could this have affected the results?

Validity of the findings

Several of the analyses miss the associated p-values within the text.
The proposed mechanisms to explain the findings are interesting, but hard to follow. A graphical representation of the findings and the proposed mechanisms, including the references, would make a great addition to the manuscript.
Please limit the conclusion statement to the significant findings.
A higher number of positive correlations were observed post intervention in the test-arm. This effect is quite pronounced based on figure 9. Is this according to expectations? What does this mean, what implications does it have?

Additional comments

Overall an interesting study, regarding set-up and results, experimental procedures and analysis seem well-conducted. I hope my suggestions will help improving the manuscript to convey the message(s) even better.

---

## Round 0.2 · Major Revisions

Dear Dr. Dhar,

Your manuscript entitled “Microbiota based personalized nutrition improves hyperglycaemia and hypertension parameters and reduces inflammation: a prospective, open label, controlled, randomized, comparative, proof of concept study" has again been carefully reviewed by the Editor and Reviewers.
Unfortunately, several of the comments raised by Reviewer 2 have not been adequately addressed. I enclose below the comments received that set out several points that will need your attention before we can consider the submission further. I urge you to pay careful attention to these points.

If you decide to resubmit a re-revised version of your manuscript, please summarize all the improvements made in the new version and answer all critical points raised in the reviewers’ report in an accompanying letter. Please copy and paste each and every reviewer's comment above your response. If you feel any of their points are inappropriate, you can provide a rebuttal in your cover letter.

Please note that resubmitting your manuscript does not guarantee eventual acceptance. I reiterate that the acceptability of the revision will depend upon the resolution of all the points raised by the reviewers.

Sincerely yours,
Stefano Menini

·

Basic reporting

A few ambiguous statements and/or overstatements are found throughout the text. Namely: - within the abstract the authors mention ‘beneficial species’ and ‘non-beneficial species’, this seems too much of a simplification. It might be that these species are pursued beneficial in the context of the current pathology, but they might not have this connotation in every context. I would therefore advice a more careful wording.
The suggestion has been addressed and the text has been modified (line 53)

 The modification of the text ‘which might have a beneficial role in the current context” forgoes the knowledge that some of these species are considered probiotic. Moreover, it is not proven at all that they are beneficial in this context. So the rephrasing of this part of the sentence is actually worse than before. The modification “which have been earlier shown to have some negative effects in the host.” in contrast, is ok.

- Lines 66 to 68 states the gut microbiota influences host nutritional status and phenotype, regulating everything from nutrient absorption, storage, metabolism to even disease development and progression. This is too strongly worded. Most likely the authors meant that the gut microbiota has an impact on a variety of processes implicated in metabolic syndrome (not syndromes, btw) and these sentences should be rephrased accordingly.
The suggestion has been addressed (line 82)

 What do the authors mean with ‘physical characteristics’ in the modified sentence? Does this refer to weight or weight gain? Because this is quite controversial.
 Syndromes is still plural.
 “the gut microbiome has been shown to adapt remarkably fast to alterations in our diet, while retaining its core composition.” It should be made clear that what changes rapidly with dietary changes is bacterial abundances and what varies little is presence/absence of bacteria. This is not clear from the sentence at this moment.

- Line 71-72: malleability of the gut microbiome is a weird expression. But more importantly, I don’t think its correct to state that gut microbial composition can be changed ‘easily’ through the diet, though easy is of course a subjective term. Microbial abundances can indeed shift rapidly upon dietary changes, but which bacteria are present or absent, is much more stable. I would advise on revising the text so that it reflects these insights.
The suggestion has been addressed (line 104-107)
 I think the line numbering is off throughout the manuscript. I don’t think this comment has been addressed sufficiently. Please see my suggestion above.

- Line 416: “modulating the other species to an even ‘and better’ distribution of abundances.” Is an even distribution ‘better’? Why would that be? And would that be the case in all circumstances?
The suggestion has been addressed and the text has been re-written (line 603)
 It’s line 472 instead of 603. The comment is not sufficiently addressed. The sentence has been changed slightly, it got a little more complicated, but it did not alter it’s meaning. The word ‘better’ is still in, without any explanation on why this would be so. I made this comment because I would like to see a more critical attitude regarding what would be an optimal evenness distribution. This likely depends on the situation and there might as well be different but equally good evenness distributions. Stating a more even distribution is ‘better’ is quite simple and might be plain wrong.


- Line 527, it is confusing to call bacteria identified within this study ‘probiotics’, even though this study (and others) indicate they have beneficial effects on health, while they were not administered to the patients. They might belong to genera that include well-known probiotics, but they were not used as such in this study. Using the term as an adjective is confusing in this context.
The suggestion has been addressed and the term “probiotics” replaced by “beneficial bacteria” (line 776).
 This is problematic. Please see my first comment.
 A suggestion: bacteria belonging to genera that include well-known probiotics. Or similar.

It is unclear how BugSpeaks works, how it comes to its recommendations, if these are reproducible, and what they were.
We have expanded the information for BugSpeaks especially in the introduction part and in the section 2.2 (line 146, line 222)
 Line numbering is off again. According to info in section 2.2: It is still not sufficiently clear how it works. It should be possible to describe this in more detail without giving away the proprietary clues if this is a problem. If it is patented, this shouldn’t be a problem. In any case, the text still does not address the reproducibility of these recommendations, and what they were.

Lines 286-289 should be part of the figure legend and not the method section.
The suggestion has been addressed

The result and discussion section would benefit from more informative subtitles that convey the message of the paragraph.
The suggestion has been addressed

It is hard to know whether the description about the significant genera covers all of those that were found significantly altered or only highlights a few interesting ones. A complete overview of all significant genera, possibly including an indication of previous findings, in table format, would be a nice addition to the manuscript.
The suggestion has been addressed. The information provided as supplementary information
 Ok. Please add a line in the text to clarify that you checked all genera and discuss x from these, as well.

Recommendations for further work are sometimes quite abruptly stated. These sentences might be improved by some introductory part identifying it as ‘recommendation for further work’. E.g. line 454.
The suggestion has been addressed and appropriate changes have been made in the text (line 667)
 Difficult to address because line numbering is off with the track changes document.

Please update the figures so that they align with each other. Specifically, convert 3D figures to 2D figures. The 3D does not add anything to their understanding. Please indicate significance (or absence thereof) between values within the figure where they are currently lacking (e.g. Fig 5; Fig 6)
The suggestion has been addressed and appropriate changes have been made
 Ok.

Experimental design

According to the exclusion criteria, antibiotic use in the two weeks prior to the study would lead to exclusion. This is a quite short time frame given that effects of antibiotics are observed for much longer timeframes. Where there any participants that took antibiotics in the last 6 months and could this have affected the results?
Point well taken. However, since we were dealing with chronic metabolic disease and not infectious disease the possibility of participants taking an antibiotic course was unlikely.
 This comment is about antibiotic intake before the study, not during the study. Participants with chronic diseases can take antibiotics for whatever reason prior to enrolment, and we know that this has an effect on microbial profiles up to 6 months after antibiotic intake. Please adapt your response accordingly.

Validity of the findings

Several of the analyses miss the associated p-values within the text.
The proposed mechanisms to explain the findings are interesting, but hard to follow. A graphical representation of the findings and the proposed mechanisms, including the references, would make a great addition to the manuscript.
We have added a new figure incorporating the suggestion (Figure 12)
 Ok.

Please limit the conclusion statement to the significant findings.
A higher number of positive correlations were observed post intervention in the test-arm. This effect is quite pronounced based on figure 9. Is this according to expectations? What does this mean, what implications does it have?
This was an observation that we thought was worth reporting. Higher positive correlations among species post intervention is a pattern that is interesting but we do not know the reason or the possible effect. We speculate that this might be due to the microbiota modulation through individualized diet. We have added relevant text in the manuscript. (line 737)
 My worry here is that it stems from a methodological issue. Can you please check and elaborate on that?

---

## Round 0.3 · accepted · Accept

Thank you for submitting a revised version of your manuscript. I am pleased to inform you that your manuscript has been accepted for publication in PeerJ in its current form.

I thank all reviewers for their efforts in improving the manuscript and the authors' cooperation throughout the review process.

Sincerely yours,
Stefano Menini